# Attenuation of Skeletal Muscle Atrophy Induced by Dexamethasone in Rats by Teaghrelin Supplementation

**DOI:** 10.3390/molecules28020688

**Published:** 2023-01-10

**Authors:** Cian-Fen Jhuo, Sheng-Kuo Hsieh, Wen-Ying Chen, Jason T. C. Tzen

**Affiliations:** 1Graduate Institute of Biotechnology, National Chung-Hsing University, Taichung 402, Taiwan; 2Department of Veterinary Medicine, National Chung-Hsing University, Taichung 402, Taiwan

**Keywords:** Akt phosphorylation, protein degradation, protein synthesis, skeletal muscle atrophy, teaghrelin

## Abstract

Muscle atrophy caused by an imbalance between the synthesis and the degradation of proteins is a syndrome commonly found in the elders. Teaghrelin, a natural compound from oolong tea, has been shown to promote cell differentiation and to inhibit dexamethasone-induced muscle atrophy in C2C12 cells. In this study, the therapeutic effects of teaghrelin on muscle atrophy were evaluated in Sprague Dawley rats treated with dexamethasone. The masses of the soleus, gastrocnemius and extensor digitorum longus muscles were reduced in dexamethasone-treated rats, and the reduction of these muscle masses was significantly attenuated when the rats were supplemented with teaghrelin. Accordingly, the level of serum creatine kinase, a marker enzyme of muscle proteolysis, was elevated in dexamethasone-treated rats, and the elevation was substantially reduced by teaghrelin supplementation. A decrease in Akt phosphorylation causing the activation of the ubiquitin–proteasome system and autophagy for protein degradation was detected in the gastrocnemius muscles of the dexamethasone-treated rats, and this signaling pathway for protein degradation was significantly inhibited by teaghrelin supplementation. Protein synthesis via the mTOR/p70S6K pathway was slowed down in the gastrocnemius muscles of the dexamethasone-treated rats and was significantly rescued after teaghrelin supplementation. Teaghrelin seemed to prevent muscle atrophy by reducing protein degradation and enhancing protein synthesis via Akt phosphorylation.

## 1. Introduction

As a consequence of the stable increase in life expectancy over the past decades, medication for age-related illnesses, such as muscle atrophy, osteoporosis and neurodegenerative diseases, has indispensably become a social responsibility all over the world [1,2]. Locomotion is important for the elderly population and relies not only on neuron regulation in the brain but also on behavior execution by the skeletal muscles. The skeletal muscles are the largest protein reservoir in a healthy human body, accounting for approximately 40–50% of total body weight and playing a crucial role in energy homeostasis [3]. It has been reported that people lose 3–8% of their muscle mass each decade after 30 years of age and more rapidly after 60 years of age [4]. Muscle atrophy leads to muscle weakness for physical performance and is commonly caused by an imbalance between protein degradation and protein synthesis, putatively resulted from many factors, such as muscle disuse, malnutrition, obesity, injury and cancer-associated cachexia [1,5,6]. Therefore, modulating the key factors involved in the degradation and synthesis of proteins is crucial for the prevention of muscle atrophy.

The enhancement of protein degradation through the ubiquitin–proteasome system and autophagy accelerates the catabolism of muscle proteins, leading to muscle wasting [5]. The rate of muscle protein degradation is in proportion to the level of serum creatine kinase (muscle type), which is regarded as a marker enzyme of muscle proteolysis [7]. A reduction in Akt phosphorylation stimulates forkhead box protein O1 (FOXO1) transcription factors, thereby inducing ubiquitin ligase proteins, Atrogin-1 and MuRF1 that are considered the hallmarks of muscle atrophy signaling. The anabolism of muscle proteins is also regulated by Akt phosphorylation; a reduction in Akt phosphorylation inhibits protein synthesis through the mTOR/p70S6K pathway in skeletal muscle [5,8,9]. Dexamethasone (Dexa), a synthesized glucocorticoid analog, has been widely used to establish animal models of muscle atrophy [5,10,11]. In animal models, intraperitoneal injection of Dexa to rats for 5 days induced muscle damage, which was implicated in disrupting the balance between protein degradation and synthesis by a reduction in Akt phosphorylation, leading to muscle atrophy [12,13].

Anabolism in elders significantly declines as a consequence of the decrease in growth hormone secretion stimulated by ghrelin, a 28-amino-acid peptide hormone with many physiological functions [14,15,16]. It has been demonstrated that ghrelin exhibited a therapeutic effect on muscle atrophy by enhancing Akt phosphorylation [17]. Teaghrelin, a unique acylated flavonoid tetraglycoside found in oolong tea, has been shown to increase appetite, enhance growth hormone secretion and display neuroprotective effects in a cellular model of Parkinson’s disease by binding to the ghrelin receptor [18,19]. Recently, teaghrelin was shown to promote cell differentiation and to inhibit Dexa-induced muscle atrophy in C2C12 cells [20]. In this study, we aimed to evaluate the therapeutic effects of teaghrelin on muscle atrophy in a Dexa-treated rat model.

## 2. Results

### 2.1. Effects of Teaghrelin on Body Weight, Skeletal Muscle Mass and Serum Creatine Kinase Level in Dexa-Treated Rats

To evaluate the effects of teaghrelin on Dexa-induced skeletal muscle atrophy, rats were supplemented daily with teaghrelin (10, 20 or 40 mg/kg) for 10 days or left untreated, and the rats in the Dexa-treated groups were injected daily with Dexa in the last five days. After Dexa administration, the body weight of the rats decreased in the last five days, and no significant effect was observed on the decrease of body weight of the Dexa-treated rats when teaghrelin was supplemented (Figure 1). The masses of the soleus, gastrocnemius and extensor digitorum longus muscles in the rats were measured and compared at the end of experiment. The results showed that the masses of all the three muscles in Dexa-treated rats were evidently lower than those in rats of the control group, and that the reduction of these muscle masses was almost prevented when the rats were supplemented with teaghrelin at 40 mg/kg (Figure 2).

Changes in the cross-sectional area of the skeletal muscle fibers were examined. The reduction of the cross-sectional area by Dexa injection in the soleus and extensor digitorum longus muscle was prevented by the supplementation of teaghrelin (Figure 3). Accordingly, the level of serum creatine kinase, a marker enzyme of muscle proteolysis, was elevated in Dexa-treated rats, and the elevation was substantially reduced when teaghrelin was supplemented (Figure 4). These results showed that teaghrelin supplementation effectively attenuated Dexa-induced muscle damage.

### 2.2. Effects of Teaghrelin on Protein Degradation in Dexa-Treated Rats

To assess the effects of teaghrelin on protein degradation in the gastrocnemius muscles of Dexa-treated rats, the phosphorylation levels of Akt and FOXO1 as well as the contents of two ubiquitin ligases (Atrogin-1 and MuRF1) and the autophagy-related protein LC-3 were measured and compared. The results showed that both Akt and FOXO1 phosphorylation levels were apparently reduced when the rats were treated with Dexa, and the reduced levels were fully recovered when 40 mg/kg of teaghrelin was supplemented (Figure 5). Accordingly, the contents of Atrogin-1, MuRF1 and LC-3 were substantially elevated when the rats were treated with Dexa, and the elevated contents were almost eliminated when 40 mg/kg of teaghrelin was supplemented (Figure 6). Taken together, the results suggested that teaghrelin supplementation prevented protein degradation by inhibiting the ubiquitin–proteasome system and autophagy through the Akt/FOXO1 pathway.

### 2.3. Effects of Teaghrelin on Protein Synthesis in Dexa-Treated Rats

To clarify the effects of teaghrelin on muscle protein synthesis in Dexa-treated rats, the phosphorylation levels of mTOR and p70S6K were measured and compared. The results showed that both phosphorylation levels (P-mTOR and P-p70S6K) were apparently reduced when the rats were treated with Dexa, and the reduced levels were mostly recovered when 40 mg/kg of teaghrelin was supplemented (Figure 7). It seemed that teaghrelin had a protective effect on protein synthesis against muscle atrophy.

## 3. Discussion

Regular exercise is the most effective strategy to maintain muscle function, but it is difficult for people who require prolonged bed rest. Therefore, preserving protein metabolism in the skeletal muscles is the primary therapy for patients with muscle atrophy [5,21,22]. Both protein degradation and synthesis were found to be regulated by Akt, a crucial modulator of muscle metabolism [23,24]. With the rising of morbidity for muscle atrophy-associated diseases, many drugs have been developed to rescue muscle atrophy; however, none of them have been found to effectively prevent the progressive loss of muscle masses in patients so far [22]. Some flavonoids have been shown to suppress Dexa-induced muscle atrophy through the Akt signaling pathway [9,25]. For example, glabridin and naringenin, were demonstrated to possess protective effects on skeletal muscle loss [26,27,28].

Quercetin is the most abundant dietary flavonoid, and its glycoside forms, shown to possess protective effects against muscle atrophy, have been included in muscle atrophy inhibitor drugs (EP3138570A1 and CA2946825A1) [29,30]. Teaghrelins in oolong tea are biosynthesized from rutin and nicotiflorin, which are glycoside derivatives of quercetin and kaempferol, respectively [31]. Similarly, teaghrelin has been shown to attenuate Dexa-induced muscle atrophy in C2C12 cells, presumably via promoting cell differentiation by increasing p38, Akt, MyoD and MHC protein expression [20]. The protective effects of teaghrelin were also observed in an animal model in this study. Nonetheless, the detailed mechanism responsible for teaghrelin’s effects and the phase of myogenesis that it targets need to be further explored.

In the current study, teaghrelin was further demonstrated to exert protective effects on skeletal muscle atrophy induced by Dexa in rats. In detail, teaghrelin effectively activated the Akt/FOXO1 pathway to inhibit the Dexa-induced effects on the ubiquitin–proteasome system (MuRF1 and Atrogin-1) and autophagy (LC-3) for protein degradation. Due to the growth hormone secretagogue receptor 1a (GHSR1a) not being expressed in skeletal muscle [32,33], the protective effects of teaghrelin were exerted possibly through other functional pathways directly or indirectly. Several studies also reported that flavonoids could suppress Dexa-induced muscle atrophy through the Akt signaling pathway [9,25]. Additionally, flavonoids could suppress GLUT4 translocation or inhibit IRS-1 signaling to reduce protein degradation when insulin resistance was elevated [34]. It remains to be investigated whether teaghrelin is also able to prevent muscle atrophy by inhibiting GLUT4 translocation or IRS-1 signaling in insulin resistance-induced animal models.

In the 10 days of our animal study, body weight was significantly decreased in the Dexa-treated rats, whereas no significant changes in body weight were observed when the rats were supplemented with teaghrelin (Figure 1). Though the decreased body weight of the rats induced by Dexa treatment could not be recovered by teaghrelin supplementation, the loss of various skeletal muscle masses in Dexa-treated rats was significantly ameliorated after teaghrelin supplementation (Figure 2). In addition, the muscle cross-sectional area, positively correlated with skeletal muscle function [35], was found reduced in Dexa-treated rats, and the reduction of the muscle cross-sectional area was rescued after teaghrelin supplementation (Figure 3).

Skeletal muscles are divided into slow-twitch and fast-twitch types, owing to their diverse fiber composition [36]. In contrast with the soleus (slow twitch) muscle and the extensor digitorum longus (fast twitch) muscle, the gastrocnemius muscle is composed of mixed (both slow- and fast-twitch) fiber types [37] and was thus selected for further examination in this study. Serum creatine kinase, a proteolysis marker of muscle atrophy progression, catalyzes a reversible reaction from creatine and adenosine triphosphate to create phosphocreatine and adenosine diphosphate [7]. As anticipated, an apparent increase in the serum creatine kinase level was detected in Dexa-treated rats, and the increased level of creatine kinase was significantly reduced after teaghrelin supplementation (Figure 4).

Either acylated or unacylated ghrelin alleviated muscle atrophy in a mouse model by binding to an unknown receptor instead of the known growth hormone secretagogue receptor [17]. Nevertheless, it was shown that ghrelin attenuated muscle atrophy by activating Akt phosphorylation [17,38,39]. SUN11031, a ghrelin agonist with a longer half-life than ghrelin, was used in a clinical trial of cachexia associated with chronic obstructive pulmonary disease and found to increase the muscle mass of patients, though no apparent improvement of muscle function was detected during the short period of observation [21]. Longer clinical trials as well as further investigation on more ghrelin agonists, particularly those from natural sources, are expected for patients suffering from muscle atrophy.

In conclusion, maintaining protein metabolism in the skeletal muscles is the most effective strategy against muscle atrophy. The present study demonstrated that teaghrelin prevented protein degradation and enhanced protein synthesis in a rat model of Dexa-induced muscle atrophy. It seems that teaghrelin supplementation is a potential therapeutic treatment for patients suffering from skeletal muscle atrophy.

## 4. Materials and Methods

### 4.1. Extraction of Teaghrelin from Shy-Jih-Chuen Oolong Tea

Shy-jih-chuen oolong tea was obtained from a local tea manufacturer in Nantou, Taiwan. Teaghrelin was extracted from Shy-jih-chuen oolong tea according to the procedure developed previously with some modification [40]. Briefly, a tea infusion was passed through Diaion HP-20 gel (Merck Millipore, Burlington, MA, USA) and eluted with 50%, 75% and 100% aqueous methanol solutions. The 75% ethanol solution fraction was collected and further purified on a Sephadex LH-20 column (Merck Millipore), eluted with 15%, 30%, 40% and 50% aqueous methanol solutions. The 50% fraction containing teaghrelin was collected and lyophilized. The yield and purity of teaghrelin were analyzed using high-performance liquid chromatography (Waters Corporation, Milford, MA, USA).

### 4.2. Animal Experiment

Male Sprague Dawley rats (6 weeks old) were purchased from BioLasco, Taiwan Co., Ltd. (Taipei, Taiwan). Dexamethasone (Dexa) was bought from Sigma-Aldrich (St. Louis, MO, USA). Muscle atrophy animal model was established by modifying the protocol developed previously [36]. The rats were housed in a controlled room maintained at a temperature of 23 ± 2 °C and humidity of 60% ± 5% with a 12 h light/dark cycle. They were given ad libitum access to tap water and a standard laboratory diet (5001 Rodent LabDiet, St. Louis, MO, USA). After a week of acclimatization, the rats were randomly divided into six groups: (i) control (Con), (ii) teaghrelin (40 mg/kg body weight), (iii) Dexa (800 μg/kg body weight), (iv) Dexa + teaghrelin (40 mg/kg body weight), (v) Dexa + teaghrelin (20 mg/kg body weight) and (vi) Dexa + teaghrelin (10 mg/kg body weight). In the following 10 days of the animal study, the rats were orally supplemented with teaghrelin (40, 20 or 10 mg/kg body weight) dissolved in saline, whereas the rats in the control and Dexa-alone groups were orally supplemented with saline. The dosages of teaghrelin used in this 10-day animal study against muscle atrophy were designed based on the observation of a previous study showing that rats supplemented with 2.5 or 7.5 mg/kg of teaghrelin showed physiological effects similar to those induced by ghrelin [18]. On the fifth day, the Dexa-treated groups received an intraperitoneal injection of Dexa (800 μg/kg body weight) for 5 consecutive days, whereas the control and teaghrelin-alone groups received a saline injection. During the 10 days of the animal study, the body weight of each rat was recorded daily prior to teaghrelin supplementation and Dexa injection. The dosages of teaghrelin used in this animal study were selected according to the dosages of other flavonoid compounds used in Dexa-induced muscle atrophy reported earlier [37,38]. The animal experiment executed in this study was approved by the Institutional Animal Care and Use Committee of the National Chung-Hsing University (IACUC Approval No: 108-026).

### 4.3. Histological Analysis

The soleus and extensor digitorum longus muscles were collected and embedded in paraffin after immersion in 10% formalin for 24 h at the end of the experiment. The muscle histology was observed under a light microscope (Olympus, BX43, Tokyo, Japan), and images were captured by a digital camera (Canon EOS 600 D, Tokyo, Japan). The paraffin blocks were cut into slices of 20 μm and stained with a hematoxylin and eosin solution for 10 min. The cross-sectional area of 100 muscle fibers from each rat was measured using the ImageJ software (National Institutes of Health, Stapleton, NY, USA).

### 4.4. Detection of Serum Creatine Kinase

At the end of the animal experiment, blood was collected from the rats, and serum samples were obtained by centrifugation (1000× *g* for 10 min) upon clotting. The creatine kinase levels in the serum samples were detected using a rat CKM (creatine kinase muscle-type) enzyme-linked immunoassay (ELISA) kit (Wuhan Fine Biotech, Wuhan, China) according to the manufacturer’s instructions.

### 4.5. Western Blotting

Gastrocnemius muscle lysates from the rats in the various groups were prepared through homogenization in cold Tissue Protein Extraction Reagent (T-PER) buffer (Thermo-Fisher Scientific, Waltham, MA, USA) containing 1% phosphatase and protease inhibitor cocktails (Merck Millipore). After homogenization, the protein lysates were centrifuged at 12,000 rpm for 10 min at 4 °C, and the supernatants were collected for the following protein detection by Western blotting. The total protein content was determined using the Bradford protein assay (Bio-Rad Laboratories, Irvine, CA, USA). The protein samples were separated on 10% or 12.5% sodium dodecyl sulfate–polyacrylamide gels by electrophoresis and then transferred onto 0.22 μm polyvinylidene fluoride membranes (Merck Millipore). The membranes were soaked in BlockPRO blocking buffer (Energenesis Biomedical, Taipei, Taiwan) for 1 h at room temperature. They were then incubated with the following primary antibodies overnight at 4 °C: anti-AKT, -P-AKT, -FOXO1, -P-FOXO1, -MuRF1, -Atrogin-1, -LC3, -mTOR, -P-mTOR, -70S6K, -P-70S6K and -Glyceraldehyde 3-phosphate dehydrogenase (all purchased from Cell Signaling Technology, Boston, MA, USA). On the following day, the membranes were washed three times with Phosphate-Buffered Saline (PBS) and then incubated with the corresponding secondary antibodies for 1 h at room temperature according to the manufacturer’s instructions for the primary antibodies. After several washes with PBS, proteins were detected using an enhanced chemiluminescence reagent (Thermo-Fisher Scientific) with the MiniChemi I system (Beijing Sage Creation Science, Beijing, China) and then quantified using ImageJ software (National Institutes of Health).

### 4.6. Statistical Analysis

All data are presented as the mean ± standard deviation (SD). Statistical analyses and the analysis of variance (ANOVA) followed by Duncan’s post hoc multiple comparison test, were performed using SigmaPlot 12.0 software (Systat, Chicago, IL, USA), and *p* < 0.05 was considered statistically significant.

## Figures and Tables

**Figure 1 molecules-28-00688-f001:**
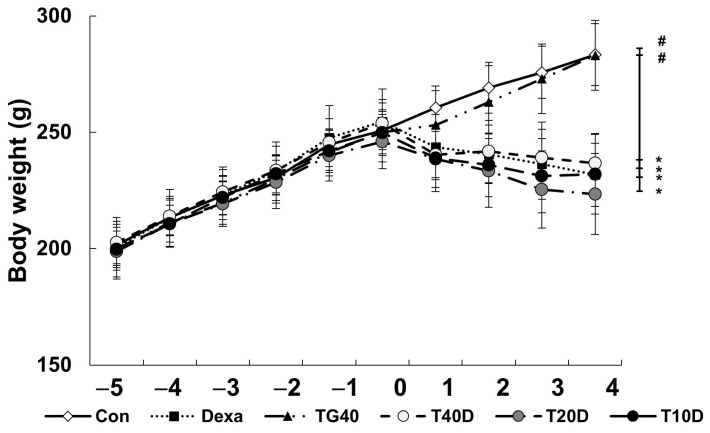
Body weight of rats before and after Dexa administration. The rats were supplemented with teaghrelin at 40 mg/kg (TG40) or with saline (Con) for 10 days, and Dexa was injected into the rats in the last five days, with the supplementation of teaghrelin (0, 10, 20 or 40 mg/kg, Dexa, T10D, T20D, or T40D) for 10 days. During the experimental period, the body weight of the rats was measured daily. The results were analyzed using one-way ANOVA. Values are presented as the mean ± SD (n = 5 or 6). * *p* < 0.05 versus Con; # *p* < 0.05 versus Dexa.

**Figure 2 molecules-28-00688-f002:**
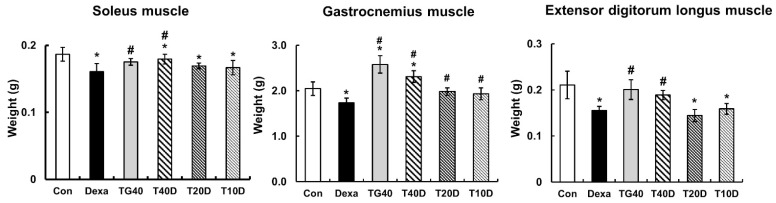
Effects of teaghrelin on the muscle masses of Dexa-treated rats. At the end of experiment, the soleus, gastrocnemius and extensor digitorum longus muscles were obtained from the rats and weighed. The Con, Dexa, TG40, T10D, T20D and T40D groups are defined in Figure 1. The results were analyzed using one-way ANOVA. Values are presented as the mean ± SD (n = 5 or 6). * *p* < 0.05 versus Con; # *p* < 0.05 versus Dexa.

**Figure 3 molecules-28-00688-f003:**
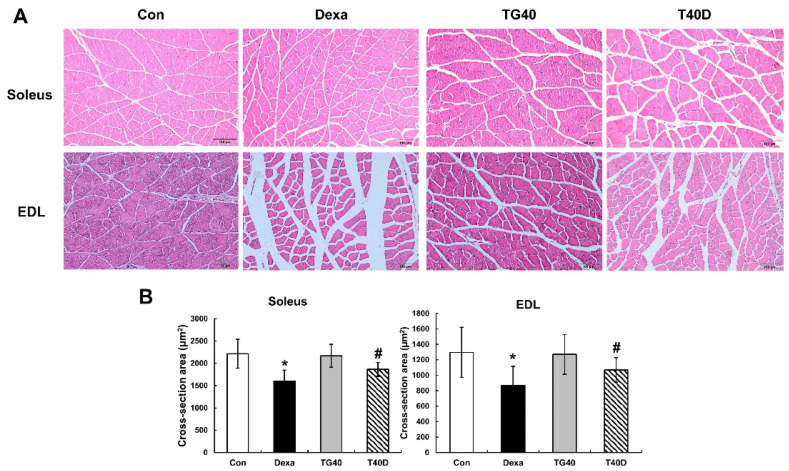
Effects of teaghrelin on muscle cross-sectional area (CSA) in Dexa-treated rats. At the end of the experiment, the soleus and extensor digitorum longus (EDL) muscles were collected. Skeletal muscle sections of the soleus and EDL were stained (hematoxylin and eosin) and observed by light microscopy (400×). Scale bar: 200 μm (**A**). The CSA of the soleus and EDL were calculated by imageJ software (**B**). The Con, Dexa, TG40 and T40D groups are defined in Figure 1. The results were analyzed using one-way ANOVA. Values are presented as the mean ± SD (n = 5 or 6). * *p* < 0.05 versus Con; # *p* < 0.05 versus Dexa.

**Figure 4 molecules-28-00688-f004:**
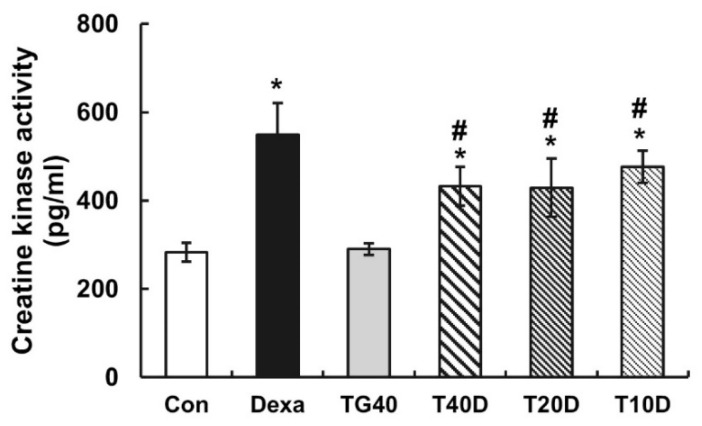
Effects of teaghrelin on the levels of serum creatine kinase in Dexa-treated rats. At the end of experiment, the levels of serum creatine kinase in the blood of the rats were measured by using ELISA. The Con, Dexa, TG40, T10D, T20D and T40D groups are defined in Figure 1. The results were analyzed using one-way ANOVA. Values are presented as the mean ± SD (n = 5 or 6). * *p* < 0.05 versus Con; # *p* < 0.05 versus Dexa.

**Figure 5 molecules-28-00688-f005:**
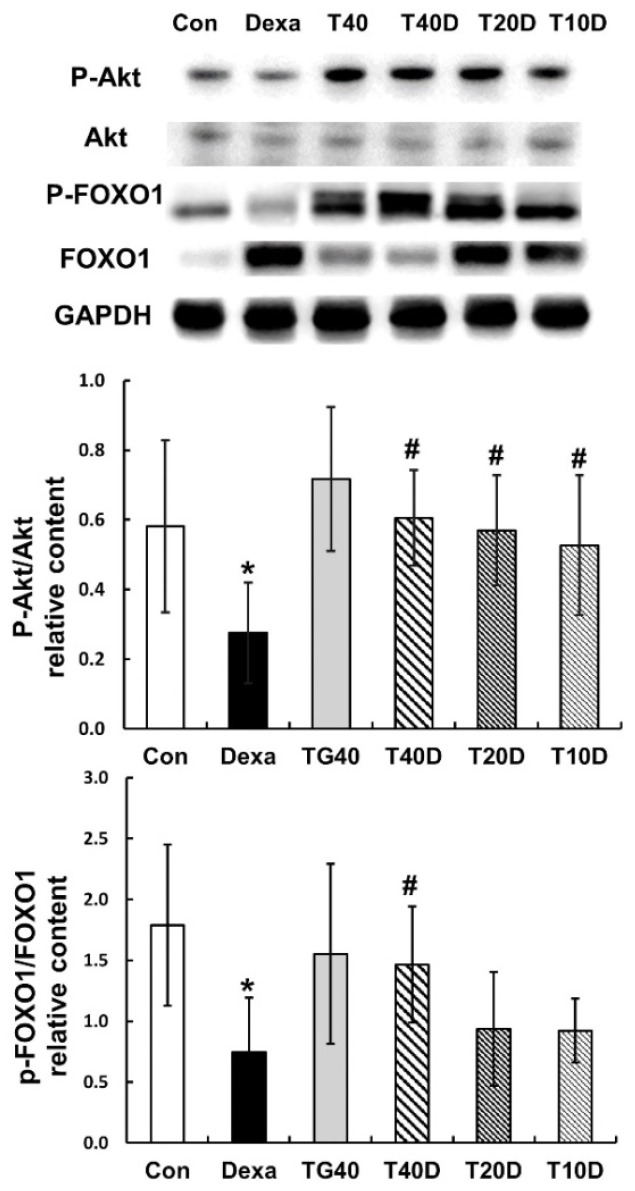
Effects of teaghrelin on Akt/FOXO1 phosphorylation in the gastrocnemius muscles of Dexa-treated rats. At the end of experiment, the phosphorylation levels of Akt and FOXO1 were measured in the gastrocnemius muscles of Dexa-treated rats by Western blotting with antibodies against Akt, P-Akt, FOXO1 and P-FOXO1. GAPDH was used as a loading control. The Con, Dexa, TG40, T10D, T20D and T40D groups are defined in Figure 1. The results were analyzed using one-way ANOVA. Values are presented as the mean ± SD (n = 5 or 6). * *p* < 0.05 versus Con; # *p* < 0.05 versus Dexa.

**Figure 6 molecules-28-00688-f006:**
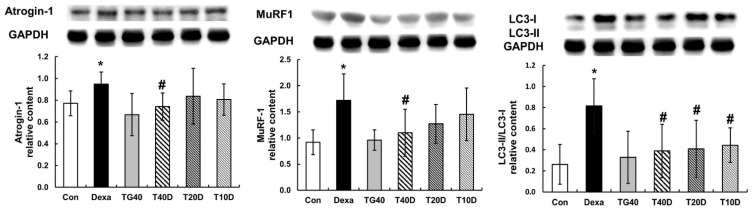
Effects of teaghrelin on three major proteins involved in muscle protein degradation in Dexa-treated rats. At the end of experiment, the contents of Atrogin-1, MuRF1 and LC-3 were measured in the gastrocnemius muscles of Dexa-treated rats by Western blotting. GAPDH was used as a loading control. The Con, Dexa, TG40, T10D, T20D and T40D groups are defined in Figure 1. The results were analyzed using one-way ANOVA. Values are presented as the mean ± SD (n = 5 or 6). * *p* < 0.05 versus Con; # *p* < 0.05 versus Dexa.

**Figure 7 molecules-28-00688-f007:**
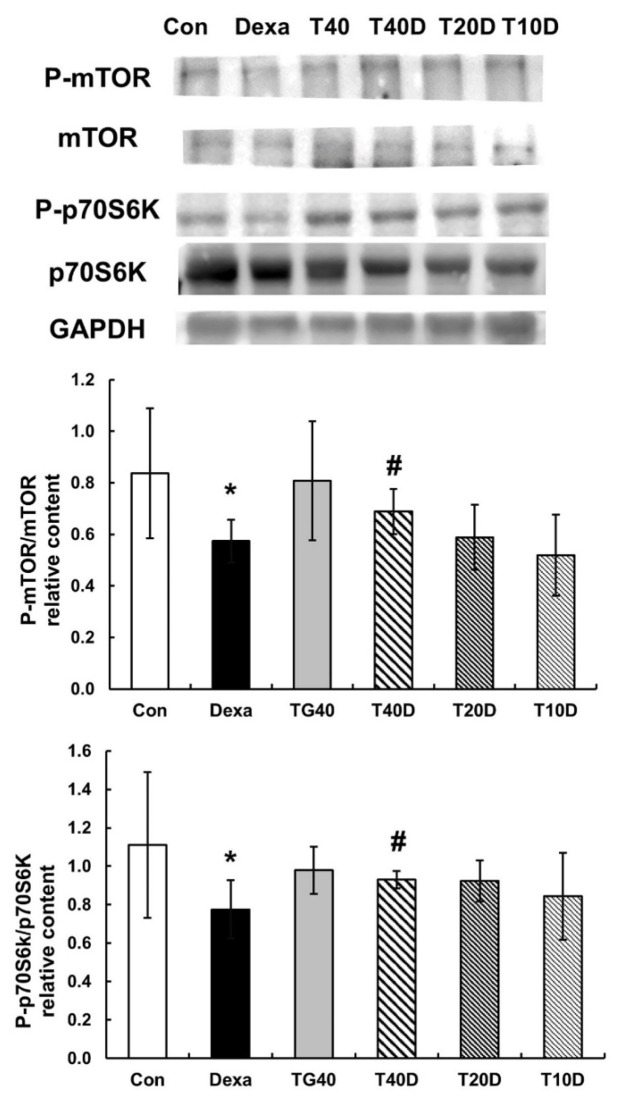
Effects of teaghrelin on mTOR/p70S6K phosphorylation in the gastrocnemius muscles of Dexa-treated rats. At the end of experiment, the phosphorylation levels of mTOR and p70S6K (P-mTOR and P-p70S6K) were measured in the gastrocnemius muscles of Dexa-treated rats by Western blotting with antibodies against P-mTOR, mTOR, P-p70S6K, p70S6K. GAPDH was used as a loading control. The Con, Dexa, TG40, T10D, T20D and T40D groups are defined in Figure 1. The results were analyzed using one-way ANOVA. Values are presented as the mean ± SD (n = 5 or 6). * *p* < 0.05 versus Con; # *p* < 0.05 versus Dexa.

## Data Availability

Not applicable.

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
