# Peer review of "Attenuation of Skeletal Muscle Atrophy Induced by Dexamethasone in Rats by Teaghrelin Supplementation"

_molecules, 2023, doi:10.3390/molecules28020688_

Round 1

Reviewer 1 Report

Cian-Fen Jhuo et al report that Attenuation of skeletal muscle atrophy induced by dexamethasone in rats by teaghrelin supplementation. This study is potentially interesting. However, there are several concerns listed below. This paper will be strengthened by addressing the following issues.

1.    The choice of concentration (teaghrelin) and their therapeutic relevance must be justified.

2.    I wonder whether teaghrelin’s effects on skeletal muscle atrophy is ghrelin receptor dependent.

3.    I wonder what is the most important regulator in the effects of teaghrelin? (among the so many targets, ex, creatine kinase, FOXO1, mTOR…) And the authors need to include inhibition study for the important regulator. If so, the authors could suggest solid mechanism for teaghrelin. Because this study is animal study, it is not easy to use TG or KO mice. If so, they should use skeletal muscle cell with knock down and overexpression technique.

Reviewer 2 Report

This describes the attenuation of skeletal muscle atrophy induced by dexamethasone in rats by teaghrelin supplementation by Jhuo et al. This is a valuable and compact, well-written  study in this field. However, some points need clarifying and certain statements require further justification.

1)As for introduction, the authors should mention in vivo models and mechanisms of muscle atrophy.

2)As for discussion, the mechanism of muscle atrophy mechanisms including the data in this study, should be mentioned from a broader perspective.

3)The authors may mention the mechanisims of teaghrelin, i.e.effects of teaghrelin on each phase such as early differentiation, and fusion.

3)It would be better if some extra data were added. i.e. histological change.

Round 2

Reviewer 1 Report

I support this work is suitable for publication in Molecules.